# Vibration as a New Survey Method for Spiders

**DOI:** 10.3390/ani14162307

**Published:** 2024-08-08

**Authors:** Rachael Harris, Robert Raven, Andrew Maxwell, Peter J. Murray

**Affiliations:** 1School of Agriculture and Environmental Science, University of Southern Queensland, Toowoomba, QLD 4350, Australia; rachael.harris@unisq.edu.au (R.H.); robert.raven@unisq.edu.au (R.R.); 2School of Engineering, University of Southern Queensland, Toowoomba, QLD 4350, Australia; andrew.maxwell@unisq.edu.au

**Keywords:** vibration, spider, survey, method, comparison, diversity index, species richness

## Abstract

**Simple Summary:**

Spiders play crucial roles in ecosystems as predators, prey, and indicators of environmental health, yet they have not been well-studied. This study compared different methods for surveying spider populations to understand their effectiveness and biases. A new vibration-based method, using an idling diesel tractor to attract spiders, was compared with the traditional methods of pitfall trapping and hand collection of spiders at night. Night collections yielded the highest species richness and diversity compared to the other methods. Pitfall traps were biased towards ground-dwelling species, while night collections targeted spiders in different vegetative strata. The study highlighted the importance of combining survey methods to accurately determine diversity in spider populations and emphasized the need for further research to better understand spider ecology. As the night collection and vibration-based methods were similar in labor required and material costs, we recommended a combination of these methods be used as there were species of spiders captured using the vibration-based method that were not captured in night collections. Further research is needed to refine the vibration-based method to better understand the underlying mechanism of how vibration attracts spiders and to improve the portability of the vibration source.

**Abstract:**

Spiders have important ecological roles as generalist predators, are a significant source of food for many other species, and are bioindicators of environmental health. However, spiders are poorly studied. Given their importance, a comparison of spider survey methods used to determine differences in spider diversity and abundance is required to understand their limitations and biases. A new survey method to attract spiders, based on vibration from an idling diesel tractor, was tested and compared to the traditional methods of pitfall trapping and hand collection of spiders at night. Across the three survey methods, there were, in total, 2294 spiders in 34 families, 138 genera, and 226 species identified. Spider species diversity and richness were significantly greater for spiders collected at night than from the other two methods (spiders collected in pitfall traps and attracted to vibration). The collection of spiders using the night collection and vibration-based methods were very similar in terms of labor required and material costs. Of all spider species identified, 80% were captured during hand collection, 30% through pitfall trapping, and 30% from vibration-based collection. Most species of spiders caught in pitfall traps were species known to be primarily ground-dwelling, whereas both arboreal and ground-dwelling spiders were collected at night and as a result of being attracted and collected using the vibration-based method.

## 1. Introduction

Spiders are ecologically very important as generalist predators [1,2]. In most locations where spiders are present, they are high in diversity and abundance and thus are able to fulfill various ecological niches and exhibit high microhabitat endemicity [1,2,3]. Spiders are excellent bioindicators when measuring the sustainability of an environment for conservation management and can be used to prioritize conservation efforts by means of spatial comparisons of site values [3,4,5]. Bioindicators reflect the health of an ecosystem when monitored by acting as early indicators of stress or loss of taxonomic diversity [5]. High sensitivity to environmental changes, such as changes to vegetation complexity, litter depth, and microclimate characteristics, means that spiders are valuable animals that can provide information on the quality and health of the environment [5,6]. Their high sensitivity as bioindicators is due to a high diversity of spiders that fill a range of environmental niches dependent on vegetation structure [3,4].

Different sampling methods for spiders can result in a sampling bias across taxa via targeting specific behaviors of spiders or vegetative structures [7,8,9]. Traditional survey methods for spiders include pitfall trapping, hand collection, Berlese funnel sampling, Malaise traps, and sweeping [7,8,10,11,12,13,14]. Berlese funnel sampling, Malaise traps, and sweeping do not accurately represent spider species richness and diversity [7,10,12,15,16,17].

Pitfall traps are the most commonly used sampling method for small ground-active fauna and, more specifically, arthropods [10,18]. Pitfall traps are useful for determining species richness and distribution to ascertain biodiversity information, as spiders caught in pitfall traps are found to have strong spatial and temporal patterns [7,10,18]. However, pitfall traps are not reliable in determining species density [10]. Pitfall traps are not time efficient (i.e., they are typically left for weeks or months to capture animals with a delay between setup and collection) and primarily only target ground-dwelling spiders. However, they are a cost-effective and passive trap that is preferentially used to survey spiders but have the limitation that they capture and kill other ground-dwelling animals [9,18].

Hand collection using the visual search method for arthropods can be undertaken during the day or at night. As it is a visual search, results from this method can be biased to species easily seen primarily through eye shine, web size, decoration, and height (within 2 m high), and spider coloration and movement [8,19]. Other factors, such as the time taken to collect the spiders and experience and knowledge of microhabitats favored by specific spider species, can also bias results [8,18,19]. As this method involves capturing species from both ground and arboreal settings, a wider range of species can be captured, including those that may be important indicator species [8,19]. The night collection of spiders has health and safety concerns associated with working at night, with the possibility of being exposed to other potentially dangerous animals and environmental conditions [20].

The use of ground-based vibration to attract spiders (and other invertebrates) is in the preliminary stages as a survey method. This method originated from anecdotal observations of many spiders being found on idling diesel tractors parked in fields during the day, and subsequently, a range of idling ‘transportable’ diesel engines have been found to attract spiders. Even though the neurobiology of how spiders interpret the vibratory cues is not currently understood, it is thought that the behavioral reaction from a vibration source is elicited from the same part of the brain that controls the predatory response [21].

In this study, we compared two traditional survey methods, pitfall traps, and nocturnal hand collection, to a vibration-based method to determine any differences in spider diversity and abundance associated with these three methods. We were also interested in understanding their respective limitations and biases (e.g., differences in labor requirements and cost).

## 2. Materials and Methods

### 2.1. Study Area

The research was undertaken within the Karrawatha Flinders Corridor, on Stewartdale, a 1200-ha property located 46 km southwest of Brisbane. Karrawatha Flinders Corridor is a 60 km stretch of open eucalypt forest with ironbark (*E. sideroxylon*), grey gum (*E. punctata*), and blackbutt (*E. pilularis*) being the most dominant species. However, more specifically, Stewartdale contains regrowth and remnant dry sclerophyll forest [22]. In open areas, the property is dominated by grass species such as *Setaria sphacelate* and *Chloris gayana* [22]. No studies on spider diversity have been undertaken within this important landscape, and as this property is managed for its conservation value for other species, e.g., koalas, having knowledge of the spider species present is valuable information for their management.

### 2.2. Sampling Methods

All spiders collected were stored in labeled, 50 mL yellow plastic screw cap specimen containers containing 70% ethanol. Surveying for spiders using different methods was conducted in Spring from 2 September to 21 October 2020. The total rainfall in this period was 8 mm. The mean monthly minimum and maximum temperatures for September were 10.3 °C to 27.5 °C and October 13.0 °C to 29.7 °C [23].

In four locations, eight similar 900 m^2^ sites (30 × 30 m) in open dry sclerophyll woodland were used; four sites (A) incorporated the use of pitfall traps, nocturnal hand collections of spiders, and the vibration-based method, and in four adjacent sites (B) only the vibration-based method was used to attract and then collect spiders (Figure 1). These four sites were named RH, RL, DR1, and DR2, each with collection sites A and B. Pairs of sites were studied to determine how many species were not captured using traditional techniques (pitfall traps and hand collection) and if the same species (with similar abundance) of spiders were captured just using the vibration-based method.

#### 2.2.1. Night Collections of Spiders

The 900 m^2^ collection areas, delineated by white reflective tape tied to trees on the boundaries, were used to conduct nocturnal hand collections of spiders for an hour once a fortnight for three consecutive fortnights in site A. The night collections were split into two 30-min intervals. For the first 30 min, two people collected spiders found above knee height, while another person focused on collecting spiders from vegetation below knee height and within the leaf litter on the ground. After this period, roles were exchanged, and the collection of spiders continued for an additional 30 min. Spiders were collected into 50 mL yellow screw cap specimen containers with a label for each site. These visual collections were undertaken at night as many species of spiders are night active and therefore much more visible at night, and visual collections of spiders were not possible during the day as we were busy collecting and processing spiders that were attracted to the vibration from the tractor.

#### 2.2.2. Pitfall Traps Collection of Spiders

Six 600 mL, 6 cm diameter plastic pitfall traps containing 100 mL of propylene glycol were placed at each of the four A sites, outside of the 900 m^2^ area, 5 m apart in two rows, starting at the back corner of the site. Each pitfall trap had a shelter placed above it to prevent entry or disruption to the pitfall trap by rain, reptiles, amphibians, or mammals. These shelters were made from a face-down plastic plate and three skewers placed evenly apart. Spiders captured in these pitfall traps were collected every fortnight on the day of the nocturnal spider collections, with a total of 1008 trap nights. Each pitfall trap was emptied into another 600 mL container, and the pitfall trap was reset with the lid and shelter both back in place.

#### 2.2.3. Vibration-Based Collection of Spiders

Vegetation was cleared in an area large enough to include the tractor with a 1 m strip in front of sites A and B for the vibration-based method (Figure 1) to ensure visibility of any spiders attracted to the vibration. This survey method used vibration from a John Deere tractor (model 6520 SE, Deere and Company, Moline, IL, USA ),idling at 750–800 rpm. The vibration from the tractor emitted a G#maj13/C chord; a middle C, which has a frequency of about 261 Hz. The tractor, with its engine idling, was located in the cleared area for one hour between midday and dusk. Spiders were only captured if they were moving towards the tractor, and only once the spiders were in the cleared 1 m wide area between the tractor and the start of site A or B. This was repeated at the front of each of the eight sites (A and B), and spiders attracted to the tractor were collected for an hour. Spiders collected were placed into a 50 mL yellow screw cap container containing 70% ethanol.

Six measurements of the vibration, emitted 1 to 40 m from the tractor, were recorded. Airborne acoustic measurements were captured using a Sony video camera (HXR-MC1500, Sony Corporation, Beijing, China), which utilized phantom-powered stereo shotgun microphones (ECM-PS1 type). An additional Zoom H2N surround sound recorder (4 microphones, Zoom Corporation, Tokyo, Japan) was used. Ground-based data were captured using ‘Audio Technica’-type surface-effect microphones contained within a custom omnidirectional resonant tube that could be driven into the ground (test point). Additionally, a wide 3″ flat metallic blade with a handle, to which a custom piezo-based pickup was attached, was used. This pickup could be driven into the ground by hand and provided directional coupling to the vibration signal. Signals were captured by a DigiDesign (Avid) MBox 2 USB2-based audio digital interface, which incorporated preamplifiers and signal padding to ensure levels did not clip nor distort and to maintain impedances of the signal lines. Data were captured using Audacity (www.audacityteam.org (version 2.4. accessed on 1 July 2020), which operates on a dedicated MacBook Pro for storage and analysis with a stock operating system installed. MatLab (tm) was then used to refine the analysis of the vibration signals. This data capture “rig” was entirely battery-powered and hence portable to move within the test site. A tape measure was used to determine localized measurement distances, and where necessary, a laser rangefinder was used for a wider range.

An ethical exemption to collect spiders was approved by the University of Southern Queensland Animal Ethics Committee (exemption ID 20EXE004).

### 2.3. Identification

Spiders were removed from their specimen containers and placed in a 100 mm petri dish with 70% ethanol under a Nikon dissection microscope, and the spider was identified using 10× magnification. Whether the spider was male, female, or juvenile was recorded. The size of each spider (tiny: <2 mm; small: 2–6 mm; medium: >6–10 mm; and large: >10 mm) was then recorded based on the length of the combined cephalothorax and abdomen. Based on its taxonomy, each spider was then classified as most commonly occurring in one of three different vegetation strata—low (on the ground or in the leaf litter), medium (above the leaf litter up to 0.5 m), and high (in vegetation above 0.5 m). Photographs were taken of the dorsal and ventral sides of each spider and recorded for later reference. For the pitfall trap collection, spiders were kept in separate containers for each pitfall trap at each site and labeled accordingly. These procedures were repeated for each pitfall trap at each site. For night collections, specimens were kept in separate containers for each site and labeled accordingly. For the vibration-based method, specimens were collected in separate containers for each 10-min interval at each of the eight sites and labeled accordingly. These processes were repeated for each site for both night collections, and spiders were captured using the vibration-based method. Spiders of all instars were identified by Dr. Robert Raven, who has over 40 years of experience as a professional arachnologist. The young of different ages were linked by a sequence from very young to adult. A placeholder name was used for the species that could not be identified at that time in the format of the first three letters of the genus followed by a number that represented individual species, e.g., *Habronestes* sp. 1 was written as *Habronestes* hab1. This designation does not denote or suggest a new species unknown to science.

### 2.4. Statistical Analyses

The data were analyzed using a two-way ANOVA with model terms for the site and trapping methods to determine the species richness and diversity (Shannon’s diversity index and Simpson’s diversity index) of spiders captured by all methods using R (version 4.0.5). R packages used include: ‘readxl’, ‘ggplot2’, ‘emmeans’, ‘multcomp’, ‘vegan’, and ‘tidyverse’. The analyses compared the results of spiders collected from pitfall traps, night collection, and during the vibration method for site A. We also compared the spiders collected at sites A and B using the vibration-based method. Means were compared using pairwise t-tests when a significant effect of the survey method was found. A probability of less than 0.05 was considered significant. Significance was expressed in different superscript groupings (‘^a^’ or ‘^b^’) with a pooled SEM and confidence level under each table. The labor required and cost in Australian dollars for each method were calculated by using material costs sourced from Bunnings (https://www.bunnings.com.au/ (accessed 6 June 2023)) and labor costs from Seek [24].

## 3. Results

Spiders collected (*N* = 2294) at Stewartdale from the four study locations were identified into 34 families, 138 genera, and 226 species (the complete list of species is in Table A1). Overall, the most diverse families in terms of the number of species were Araneidae (41 species), followed by Salticidae (37 species), Theridiidae (30 species), Gnaphosidae (19 species) and Corinnidae (15 species). Approximately 9% of spider species were captured using all three methods, with 68% captured using only one method (night collection 51%, pitfall traps 9%, and vibration-based 8%) (Figure 2).

There were highly statistically significant differences (Table 1) between survey methods for the Shannon diversity index (F2,6 = 20.916, *p* < 0.001) and species richness (F2,6 = 47.026, *p* < 0.001), and a statistically significant difference for the Simpson’s diversity index (F2,6 = 6.077, *p* = 0.036). There were no significant differences between the locations DR1, DR2, RH, and RL for species richness (F3,6 = 0.648, *p* = 0.612), diversity for the Shannon diversity index (F3,6 = 0.192, *p* = 0.897), and Simpson’s diversity index (F3,6 = 1.13, *p* = 0.409) of spiders captured by the different methods.

There was no significant difference in spider species richness between sites A and B when spiders were collected using the vibration-based method (F1,3 = 2.298, *p* = 0.227) (Table 2). There were no significant differences between sites A and B for spider diversity when calculated using the Shannon diversity index (F1,3 = 0.086, *p* = 0.788) or Simpson’s diversity index (F1,3 = 0.632, *p* = 0.485) (Table 2).

Night collections of spiders had the highest percentage of taxa in both the number of species and families of spiders. The pitfall trap and vibration-based method resulted in the same number of species of spiders; however, pitfall traps captured a greater number of families than those collected from the vibration-based method (Figure 3).

Night collections and vibration-based collections of spiders were similar in total time required for preparation, spider collection and to reset traps per one collection location (Table 3). These two methods were also similar, with the total overall cost between AUD 100.75 and AUD 180.75 for both materials and personnel labor [24], whilst pitfall traps had the highest running cost of AUD 215 per one collection location.

The vibration-based method and pitfall traps both primarily targeted spiders from the low vegetative stratum with 85% of spider species normally found on the ground or in leaf litter, with the remaining 15% from the middle or high vegetative strata (Figure 4).

Collection of spiders at night had the highest percentage of species found in high or middle vegetative strata (63%) whilst the remaining 37% were found in the low vegetative stratum (Figure 4).

The five most abundant spider species collected were different for each survey method. *Habronestes* “hab4” (Zodariidae) was the only common species caught using all three survey methods, with 55 caught during the three night collections, 14 caught in six weeks of pitfall trapping, and 80 captured during the use of the vibration-based method., *Argiope keyserlingi* (Araneidae; orb-weaving spiders) was the most abundant species caught during night collections (*n* = 90), whilst Genus M sp.1 (Lycosidae) (*n* = 37) and *Habronestes* “hab2” (Zodariidae) (*n* = 98) (both ground-dwelling families) were the spiders most captured in pitfall traps and from the vibration-based method (Figure 4).

The three most abundant spider families for each survey method were different (Figure 4). Zodariidae was the only family common to all three survey methods, with 126 spiders from night collections, 41 spiders in pitfall traps, and 225 spiders collected during the use of the vibration-based method (Figure 4). Night collections resulted in the greatest number of Araneidae and Theridiidae spiders, whilst the use of pitfall traps resulted in the greatest number of Lycosidae and Gnaphosidae spiders, and the vibration-based method had the greatest number of Zodariidae and Miturgidae spiders (Figure 4).

## 4. Discussion

The eight study sites at Stewartdale contained a highly diverse spider fauna, including an estimated four previously undescribed species, highlighting how poorly studied spiders are and emphasizing the need to optimize the efficiency of spider survey methods [6,7,10,11,13,16,18]. As demonstrated in this study, the survey methods used can greatly influence the spider fauna captured through collection biases such as targeted vegetative strata, length or duration of the method, or specific spider behaviors [7,8,9,25].

### 4.1. Methodology Efficiency

An efficient and cost-effective survey method minimizes preparation time, labor, and OH&S risks and reduces material costs [26]. An effective survey technique should be quick to perform while yielding the most desirable results, such as the highest species richness, diversity, or abundance of a specific spider family [26]. The vibration-based method and night collection were more cost-effective (for material and personnel) and time-efficient than pitfall traps (Table 3). The collection time for both the vibration-based technique and night collection was one hour. However, night collections were more intensive and posed higher OH&S risks compared to the vibration-based technique or pitfall traps (Table 3) [14]. The costs for night collection and vibration-based were similar, with night collection costing AUD 47 per location, excluding the essential one-off cost of a head torch. In contrast, pitfall trapping was less efficient and more costly than both night collection and the vibration-based technique [26,27]. Pitfall traps require four hours to prepare, a minimum of six weeks to collect spiders, one hour every fortnight to reset, and cost AUD 70 per location. Despite minor differences in the efficiency of night collection and the vibration-based technique, notable differences were observed in the species of spiders each method captured.

### 4.2. Species Richness and Diversity

Spider species diversity and richness were significantly different between methods used (pitfall traps, night collections, and vibration-based collections) than they were between locations. Each method collected spider species that were not collected through the use of the other two methods, and thus, all three methods contributed to the overall species richness and diversity (Figure 2). There was no significant difference between species richness or diversity for spiders collected using the vibration-based method at sites A and B (Table 2). Thus, night collections and pitfall trapping collections did not affect the species richness or species diversity caught in site A. From here, reference to the vibration-based method will only refer to the specimens caught in site A. The species and families of spiders and thus the diversity index and species richness from night collections of spiders were significantly greater than those from pitfall traps and the vibration-based method (Table 1). This could be due to the greater diversity of spider species in higher and more complex vegetative strata, which may have been easier to find. The use of pitfall traps for six weeks may have captured more spiders and thus a sampling bias compared to the vibration-based method with only a one-hour-long sampling effort [9]. Night collection had the highest percentage of taxa found from the total number of taxa across all methods (Figure 3). This method captured 80% of all the species and 85% of all the families. These results agree with other studies using night collection, where this method produced between 50% and 80% of the total number of species sampled [25,27,28]. Night collections vary with the number of people searching, the area searched, the vegetative strata targeted (e.g., above ankle height), or vegetation type (e.g., rainforest/Mediterranean forest) [25,27,28].

Both pitfall traps and the vibration-based method captured 35% of all species found; however, pitfall traps captured spiders in 65% of the families, whilst the vibration-based method captured spiders from only 55% of families found at Stewartdale (Figure 3). Thus, the vibration-based method attracted more spider species within a smaller number of families than did pitfall traps despite the difference in time (one hour compared to six weeks). In contrast, in Tasmanian heathland, pitfall traps collected a higher number of families (31) and species (113) than in the hand collection of spiders in the same landscape (23 families and 53 species) [7]. However, this study utilized a significantly larger number of pitfall traps (108 in total) across four locations [7]. Hand collection was only performed for 30 min during the day, another factor that has been suggested to influence the number of species collected [7,27] and the differences in the number of families and, in particular, species collected in the two studies.

There was an observable difference in the targeted vegetative strata across methods used to survey spiders (Figure 4). Pitfall traps and the vibration-based method target very similar vegetative strata, where 85% of the spider species were from the low vegetative stratum (on the ground or in leaf litter below 0.5 m), with the remaining 15% found in the middle or high vegetative strata (above 0.5 m) (Figure 4). Comparatively, night collections resulted in 63% of spider species collected having been located in middle or high vegetative strata, with the remaining 37% found in low vegetative strata (Figure 4). Differences are possibly due to a sampling bias from the methods used [29]. Night collections involve active searches for spiders in all vegetative strata and thus were more likely to result in a higher abundance of spiders caught in middle and high vegetative strata than in pitfall traps (Figure 4) [7,29,30,31,32]. The most abundant species collected generally belonged to the most abundant families found (Figure 4).

A high abundance of ground-active spider species in the Lycosidae, Gnaphosidae, and Zodariidae families (Figure 4) caught in pitfall traps can be attributed to their inability to escape from the pitfall traps [33]. Most spiders caught in pitfall traps do not have claw tufts that allow them to attach themselves to vertical slippery surfaces such as the plastic of the pitfall traps [21,29,33]. Differences in the most abundant families of spiders captured in pitfall traps and the vibration-based method could be attributed to claw tufts that are only present in the families Corinnidae and Miturgidae; spiders in these families were abundantly collected using the vibration-based method but not in pitfall traps. Spiders of the Corrinidae and Miturgidae families may not have been as abundant in pitfall traps as their claw tufts may have allowed them to escape falling into the pitfall traps or allowed them to climb out of the traps [34,35].

The three methods used highlight differences in the diversity and abundance of spiders collected which in part was attributed to differences in search effort, cost and behavioral differences in the spiders. Clearly, a combination of different survey methods is required as not all spider species were collected by just one of the survey methods used. These differences in diversity and abundance of spiders collected may be due in part to the timing of the surveys (spring), the spiders’ size, and the spider’s location at the beginning of the vibration. Furthermore, the vibration-based method attracted spiders not collected using the other two traditional methods, and therefore, further research is required to ascertain why this occurred (the mechanism), the effect vibration has on other animals, and answers to other questions, including what frequencies are different spider species attracted to, how far away from the vibration source are spiders attracted, and how environmental factors, e.g., substrate, moisture content, temperature might affect this distance.

## 5. Conclusions

Night collections of spiders had the highest species richness and species diversity compared to the use of pitfall traps and the vibration-based method. The vibration-based method produced more species with fewer families compared to pitfall traps that collected fewer species with a greater number of families (Table 1). Night collections contributed at least 80% of all species collected by the different methods and would be necessary for a comprehensive survey of spiders in an area. Pitfall traps and the vibration-based method overlap in the targeted vegetative strata, and if time were limited, the vibration-based method is preferred. The vibration-based method would be preferred over pitfall traps to capture families with claw tufts, such as Corinnidae and Miturgidae. Each of these methodologies targets different assemblages of spider communities, and thus, each has their advantages and disadvantages. Clearly, using one survey method does not represent the complete assemblage of spider species within an area. Future research is needed to understand the effects of varying vibration frequency on different spider species to increase survey efficiency. A more compact vibration generator needs to be developed to allow access to locations inaccessible to a tractor to survey spiders. When a smaller vibration-generating device is developed, i.e., a tractor is no longer required, the vibration-based method could be a more time and cost-effective alternative to pitfall traps but should be used in conjunction with night collections to obtain a broader, more representative spider community assemblage. Further research into vibration as a survey method is required.

## Figures and Tables

**Figure 1 animals-14-02307-f001:**
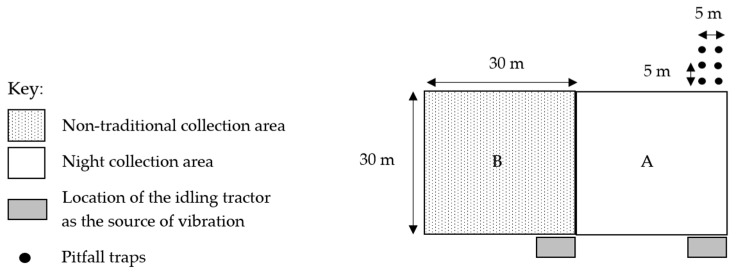
Survey method layout (A and B) replicated in four similar locations across Stewartdale, in southeast Queensland, Australia.

**Figure 2 animals-14-02307-f002:**
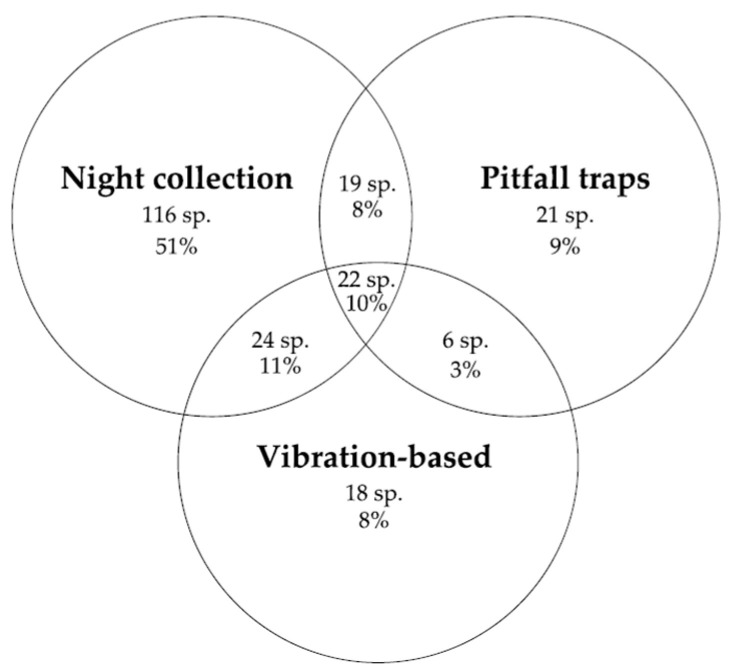
Venn diagram of the number and percentage of spider species caught using the three survey methods showing their overlap.

**Figure 3 animals-14-02307-f003:**
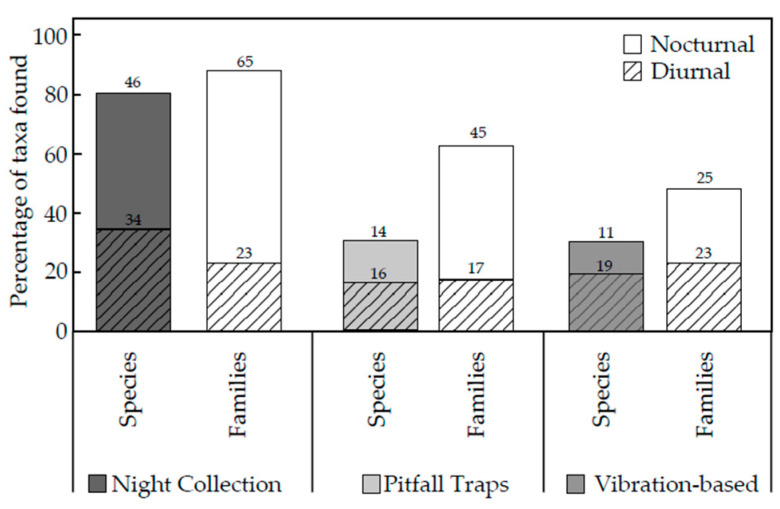
The number of species and families of spiders considered to be diurnally or nocturnally active captured by the different methods as a percentage of the total species and families recorded in the four (A) study sites.

**Figure 4 animals-14-02307-f004:**
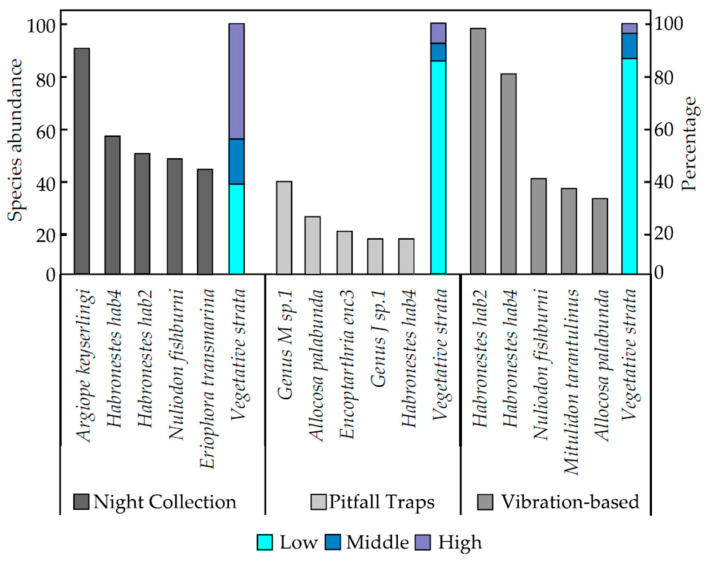
The five most abundant spider species captured from each survey method for night collection, pitfall traps, and the vibration-based method, and the percentage of spiders caught in the three different vegetative strata (low, medium, and high) for the three methods.

**Table 1 animals-14-02307-t001:** Mean values of species richness and diversity of spiders captured using night collection, vibration-based methods, and from pitfall traps across four locations on Stewartdale in southeast Queensland, Australia.

Survey Method	Shannon Diversity Index	Simpson Diversity Index	Species Richness	No. Species
Night Collection	5.84 ^a^	0.976 ^a^	92.7 ^a^	181
Pitfall Traps	4.37 ^a^	0.945 ^ab^	30.2 ^b^	68
Vibration-based site A	3.88 ^b^	0.904 ^b^	29.2 ^b^	70
Pooled SEM	0.223	0.014	7.5	
Confidence Level	0.95	0.95	0.95	

^ab^ Within columns means followed by the same superscript letter were not significantly different.

**Table 2 animals-14-02307-t002:** Shannon and Simpson Diversity indices and species richness means (with standard errors) for vibration-based collections of spiders at sites A and B.

Survey Method	Shannon Diversity Index	Simpson Diversity Index	Species Richness	No. Species
Vibration-based site A	3.88	0.904	29.2	70
Vibration-based site B	3.81	0.924	20.0	43
Pooled SEM	0.167	0.018	6.8	
Confidence Level	0.95	0.95	0.95	

**Table 3 animals-14-02307-t003:** Labor and material/equipment costs of each survey method per collection location.

Parameter	Night Collection	Pitfall Traps (6)	Vibration-Based
Preparation time	1 h	4 h	2 h
	Put ethanol in specimen containers	Buy materials and construct pitfall traps and lids	Put ethanol in specimen containers
	Put labels in specimen containers	Dig the pitfall holes and set the pitfall traps	Labeled specimen containers in labeled bags
	Specimen containers in labeled bags	Moving between each trap	Clearing of ground for tractor
Collection time	1 h and 15 min	1 h	1 h and 15 min
	Collection of spiders and moving between sites	Filter out the pitfall contents into separate containers	Collection of spiders and transit time between locations with tractor
		Refill pitfall traps, moving between sites	
Total cost of material	AUD 47–AUD 127	AUD 70	AUD 47
	AUD 20 specimen containers	AUD 30 propylene glycol	AUD 20 specimen containers
	AUD 2 labels	AUD 10 containers	AUD 2 labels
	AUD 10 pooter	AUD 30 accessories	AUD 10 pooter
	AUD 80 head torch (one off cost)		AUD 10 tractor fuel
	AUD 15 ethanol		AUD 15 ethanol
Labor	Intensive	Short intensive	Short intensive
Labor (h)	2.25	5	3.25
OH&S risk	High	Low	Low
	Walking through the bush at night		
Field Ecologist (AU AUD 43/h)	AUD 96.75/person	AUD 215/person	AUD 139.75/person
Total cost in AUAUD	AUD 143.75–223.75	AUD 285	AUD 186.75

This table does not include the time or cost for identification of spider species (which would be similar for each method) or the cost of renting or buying a tractor.

## Data Availability

Data are available upon reasonable request from the authors.

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
