# Peer review of "Vibration as a New Survey Method for Spiders"

_animals, 2024, doi:10.3390/ani14162307_

Round 1

Reviewer 1 Report

Comments and Suggestions for Authors

Manuscript ID: animals-3116117

Title: Vibration as a new survey method for spiders

Authors: Rachael Harris, Robert Raven, Andrew Maxwell, Peter J. Murray

Summary: This study compared three different survey techniques used to find spiders: visual search (at night), pitfall traps, and a new technique, vibration + visual search. The authors found the greatest richness and diversity using visual search at night; however, the other techniques found families and species (dominated by spiders from the lowest vegetative strata) not seen during visual search. Thus, a combination of survey techniques is necessary to sample spider diversity adequately.

General Comments: Congratulations are a well-written and interesting study. I enjoyed reading it. My suggestions are relatively simple and focus on improving clarity and consistency.

Specific Comments:

Note: I read this article from start to finish in a single sitting. Thus, my comments will have a stream-of-consciousness quality.

Line 18: The authors mention "efficiency." At this point in the article, it is unclear how "efficiency" was measured and is therefore ambiguous. I recommend adding some qualifying statements to improve efficiency and clarity.

Lines 54-56: The sentence beginning with "Traditional survey methods…" would benefit from citations. I would like to see references to surveys that employed and/or evaluated each method.

Lines 62-66: This sentence claims that "Pitfall traps are not time efficient..." However, it is not clear why. One could consider pitfall traps more efficient since they can be set and left unattended, collecting many individuals without constant monitoring. Please provide a better justification for this claim.

Lines 68-69: The authors provide some evidence for bias in visual search (eye shine and conspicuous webs). I would like to see a more thorough description of visual search bias. What about spider coloration, size, web decoration, web height, etc.

Lines 73-75: A reference is warranted for this sentence. Currently, it reads as a reasonable assumption being proposed as a fact.

Lines 76-82: This paragraph needs more development. Has spider aggregation on tractors or diesel engines been formally documented, or is this anecdote? How does a motor's frequency and intensity of vibration compare with natural signals?

Lines 83-85: The end of the introduction felt rushed. I believe that the article would benefit from a stronger statement regarding the need for alternative survey techniques. I also suggest adding a sentence or two about "in this study we…" where the study objectives are clearly stated.

Section 2.1: This section felt lackluster. Why did I need to know these details? Please either provide a stronger justification for this information or reduce it to only what is necessary. I recommend the former approach.

Has this area for forest type been surveyed previously? If so, comparing these findings with previous surveys here would be very informative in evaluating the effectiveness of vibration + visual search as an alternative or additional technique.

Lines 101-119: The paragraph describing the different survey methods took much work to understand.

                First, each method should get its own paragraph, possibly a subheading.

                Second, I would like to see a justification for searching at night and not during the day, especially as the vibration + visual search technique was conducted from afternoon to dusk.

                Third, I need clarification on how the vibration + visual search was conducted. I see where the tractor was parked and the total size of the visual search area, but I need to understand how much of the area was searched and how the 10-minute intervals factor into this. I am trying to understand why the visual search was conducted for an entire hour, but the vibration + visual was less. I also need clarification on why search effort was not standardized since search effort is an essential factor that can bias results, similar to the examples the authors provided in lines 69-71.

Lines 116-117: "using appropriate equipment"… please provide details on how these measurements were taken.

Lines 139-140: Please provide the name of the spider identifier.

Section 2.4: More detail is needed here. For example, how was diversity calculated? Was it calculated using R? What packages were used? Were assumptions tested and met? How were the data transformed? The same locations were surveyed at several time points. If so, you have a repeated measures design and a repeated measures ANOVA or a linear mixed model that is more appropriate than a two-way ANOVA. Why report Pooled SE instead of individual confidence intervals?

Results: I recommend also reporting some measure of effect like R2 and partial-eta2 or a beta value depending on the statistical method used.

Please provide actual P values instead of P < 0.05 and P > 0.05. I have no problem with P < 0.001.

Lines 186-187: While interesting, how is knowing that the tractors produce a G#maj13/C chord useful? Would not reporting the specific frequency be more helpful? I suggest reporting whatever measure makes reproducing this vibration frequency easier for others.

 Figure 3: Is there a reason why the author chose not to statistically test for differences in diurnal vs. nocturnal species by technique? Reporting the information is great, but I think that also testing it statistically would be better.

Table 3: The "total cost of materials" for night collection ranged quite a bit. Was this range due to finding more spiders than the other techniques?

Discussion: The discussion should be further developed and divided into subsections. For example, since the methods used for each survey technique differed in the search effort, how much of the differences observed can be attributed to effort vs. technique? How did the results compare to results from previous surveys, either the author's own or by others? Did the authors calculate a rarefaction curve to see how well their search efforts were doing at finding available species? What limitations does the study have, and how can the vibration + visual search be improved?

The last three paragraphs of the discussion and the conclusion were too redundant concerning the results section. The author repeated their results instead of adequately interpreting and contextualizing them. Remove repetition of the results and expand their argument for what they think their results indicate, how they are useful, and what they think should be done next. For example, how far away from the vibration source can the vibrations be detected? How far away can a spider detect them? How does substrate (air, plant, ground, etc.) affect this distance? While vibration is important for predation, it is also crucial for intersexual interactions, i.e., mate detection. However, web-building spiders likely detect and use vibration differently than free-moving species like lycosids and others. Addressing these questions and others like this would significantly improve the manuscript and address its limitations and implications more clearly and thoroughly.  

Lines 310-313: The authors report data on sex and developmental stage differences in the discussion section when it should have been reported in the results, analyzed statistically when possible, and interpreted in the discussion.

Congratulations again.

Author Response

Summary: This study compared three different survey techniques used to find spiders: visual search (at night), pitfall traps, and a new technique, vibration + visual search. The authors found the greatest richness and diversity using visual search at night; however, the other techniques found families and species (dominated by spiders from the lowest vegetative strata) not seen during visual search. Thus, a combination of survey techniques is necessary to sample spider diversity adequately.

General Comments: Congratulations are a well-written and interesting study. I enjoyed reading it. My suggestions are relatively simple and focus on improving clarity and consistency.

Specific Comments:

Note: I read this article from start to finish in a single sitting. Thus, my comments will have a stream-of-consciousness quality.

Comment 1: Line 18: The authors mention "efficiency." At this point in the article, it is unclear how "efficiency" was measured and is therefore ambiguous. I recommend adding some qualifying statements to improve efficiency and clarity.

Response 1: Efficiency has been replaced with ‘labour required’ in lines 18, 33 and 190.

Comment 2: Lines 54-56: The sentence beginning with "Traditional survey methods…" would benefit from citations. I would like to see references to surveys that employed and/or evaluated each method.

Response 2: We have added in relevant citations (7,8,10-14) for lines 54-56, for the sentence starting with “Traditional survey methods…”

Comment 3: Lines 62-66: This sentence claims that "Pitfall traps are not time efficient..." However, it is not clear why. One could consider pitfall traps more efficient since they can be set and left unattended, collecting many individuals without constant monitoring. Please provide a better justification for this claim.

Response 3: We have clarified why pitfall traps are not time efficient by adding on lines 62-66 ‘(i.e., they are typically left for weeks or months to capture animals with a delay between setup and collection’).

Comment 4: Lines 68-69: The authors provide some evidence for bias in visual search (eye shine and conspicuous webs). I would like to see a more thorough description of visual search bias. What about spider coloration, size, web decoration, web height, etc.

Response 4: Lines 68-70 have been amended to include more examples of bias from visual search ‘As it is a visual search, results from this method can be biased to species easily seen primarily through eye shine, web size, decoration and height (within 2 m high), and spider coloration and movement [8,19].’

Comment 5: Lines 73-75: A reference is warranted for this sentence. Currently, it reads as a reasonable assumption being proposed as a fact.

Response 5: A reference has been provided for lines 74-76 and is included in the References. Lamarre GPA, Juin Y, Lapied E, Le Gall P, Nakamura A (2018) Using field-based entomological research to promote awareness about forest ecosystem conservation. Nature Conservation 29: 39–56. https://doi.org/10.3897/natureconservation.29.26876

Comment 6: Lines 76-82: This paragraph needs more development. Has spider aggregation on tractors or diesel engines been formally documented, or is this anecdote? How does a motor's frequency and intensity of vibration compare with natural signals?

Response 6: Lines 78-83: The paragraph has been modified to acknowledge it is only anecdotal ‘originated from anecdotal observations’ and we do state that ‘that the behavioural reaction from a vibration source (from a tractor) is elicited from the same part of the brain that controls the predatory response [17].’  

Comment 7: Lines 83-85: The end of the introduction felt rushed. I believe that the article would benefit from a stronger statement regarding the need for alternative survey techniques. I also suggest adding a sentence or two about "in this study we…" where the study objectives are clearly stated.

Response 7: Lines 84-88 were modified and the objectives of the study added to lines 84-88 ‘In this study we compared two traditional survey methods, pitfall traps and nocturnal hand collection, to a vibration-based method to determine any differences in spider diversity and abundance associated with these three methods. We were also interested in understanding their respective limitations and biases (e.g. difference in labour requirements, cost).’

Comment 8: Section 2.1: This section felt lackluster. Why did I need to know these details? Please either provide a stronger justification for this information or reduce it to only what is necessary. I recommend the former approach.

Has this area for forest type been surveyed previously? If so, comparing these findings with previous surveys here would be very informative in evaluating the effectiveness of vibration + visual search as an alternative or additional technique.

Response 8: The following sentence has been added to Section 2.1. (lines 97-100) to address this comment. No studies on spider diversity had been undertaken within this important landscape and as this property is managed for its conservation value for other species, e.g. koalas, having knowledge of the spider species present is valuable information for their management.’

Comment 9: Lines 101-119: The paragraph describing the different survey methods took much work to understand.

                First, each method should get its own paragraph, possibly a subheading.

                Second, I would like to see a justification for searching at night and not during the day, especially as the vibration + visual search technique was conducted from afternoon to dusk.

                Third, I need clarification on how the vibration + visual search was conducted. I see where the tractor was parked and the total size of the visual search area, but I need to understand how much of the area was searched and how the 10-minute intervals factor into this. I am trying to understand why the visual search was conducted for an entire hour, but the vibration + visual was less. I also need clarification on why search effort was not standardized since search effort is an essential factor that can bias results, similar to the examples the authors provided in lines 69-71.

Response 9: Subheadings for the three methods have been included, where appropriate. As now explained in Section 2.2. visual collections were undertaken at night as many species of spiders are night active and therefore much more visible at night. Furthermore, visual collections of spiders were not possible during the day as we were busy collecting and processing spiders that were attracted to the vibration from the tractor. Detailed clarification has been provided by adding more information about the methodology used within lines 111-149. This clarification now makes it clear that both the nocturnal spider collections and the vibration-based collections were both for a total of 1 hour each time they were undertaken.

Comment 10: Lines 116-117: "using appropriate equipment"… please provide details on how these measurements were taken.

Response 10:  Lines 150-166: Additional information how vibration from the tractor was recorded has been added to the manuscript.

‘Airborne acoustic measurements were captured using a Sony video camera (HXR-MC1500) which utilised phantom powered stereo shotgun microphones (ECM-PS1 type). An additional Zoom H2N surround sound recorder (4 microphone) was used. Ground based data was captured using `Audio Technica’ type surface-effect microphones contained within a custom omnidirectional resonant tube which could be driven into the ground (test point). Additionally, a wide 3” flat metallic blade, with handle, to which a custom piezo-based pickup attached was used. This pickup could be driven into the ground by hand and provided directional coupling to the vibration signal. Signals were captured by a DigiDesign (Avid) MBox 2 USB2 based audio digital interface which incorporated preamplifiers and signal padding to ensure levels didn’t clip nor distort, and to maintain impedances of the signal lines. Data was captured using Audacity (www.audacity.com) operating on a dedicated MacBook Pro for storage and analysis with a stock operating system installed. MatLab (tm) was then used for more refined analysis of the vibration signals. This data capture “rig” was entirely battery powered and hence portable to move within the test site. A tape measure was used to determine localised measurement distances, and where necessary a laser rangefinder used for wider ranging.’

Comment 11: Lines 188-189: Please provide the name of the spider identifier.

Response 11: Our initial advice was not to name the arachnologist; clearly this was an error and we have now provided the name of the spider identifier, Dr Robert Raven, in line 189.

Comment 12: Section 2.4: More detail is needed here. For example, how was diversity calculated? Was it calculated using R? What packages were used? Were assumptions tested and met? How were the data transformed? The same locations were surveyed at several time points. If so, you have a repeated measures design and a repeated measures ANOVA or a linear mixed model that is more appropriate than a two-way ANOVA. Why report Pooled SE instead of individual confidence intervals?

Response 12: Diversity (both Shannon’s diversity index and Simpsons diversity index) was calculated using R, given in lines 196-200. R packages are reported in lines 199-200. Advise from the Faculty statistical advisor was followed in regards to the statistical analyses used. Confidence levels are given in Tables 1 and 2.

Comment 13: Results: I recommend also reporting some measure of effect like R2 and partial-eta2 or a beta value depending on the statistical method used.

Please provide actual P values instead of P < 0.05 and P > 0.05. I have no problem with P < 0.001.

Response 13: More information of the statistical methods used is now included with the actual P values where appropriate within lines 224-237.

Comment 14: Lines 186-187: While interesting, how is knowing that the tractors produce a G#maj13/C chord useful? Would not reporting the specific frequency be more helpful? I suggest reporting whatever measure makes reproducing this vibration frequency easier for others.

Response: The frequency is now included in line 143.

Comment 15: Figure 3: Is there a reason why the author chose not to statistically test for differences in diurnal vs. nocturnal species by technique? Reporting the information is great, but I think that also testing it statistically would be better.

Response: A comparison of the Night collection and the Vibration-based methods answers this question as there are significant differences between the two methods for Shannon and Simpson Diversity Indices (P<0.001) and Species richness (P<0.05) as shown in Table 1.

Comment 16: Table 3: The "total cost of materials" for night collection ranged quite a bit. Was this range due to finding more spiders than the other techniques?

Response: The range for total cost of materials was due to a one-off cost of a head torch, listed under total cost of materials in Table 3.

Comment 17: Discussion: The discussion should be further developed and divided into subsections. For example, since the methods used for each survey technique differed in the search effort, how much of the differences observed can be attributed to effort vs. technique? How did the results compare to results from previous surveys, either the author's own or by others? Did the authors calculate a rarefaction curve to see how well their search efforts were doing at finding available species? What limitations does the study have, and how can the vibration + visual search be improved?

Response 17: Significant proportions of the Discussion have been edited with additional information included. A comparison of our results with another Australian study comparing survey methods is discussed on lines 337-343. Further research required on the vibration-based survey method are discussed in lines 367-378.

Comment 18: The last three paragraphs of the discussion and the conclusion were too redundant concerning the results section. The author repeated their results instead of adequately interpreting and contextualizing them. Remove repetition of the results and expand their argument for what they think their results indicate, how they are useful, and what they think should be done next. For example, how far away from the vibration source can the vibrations be detected? How far away can a spider detect them? How does substrate (air, plant, ground, etc.) affect this distance? While vibration is important for predation, it is also crucial for intersexual interactions, i.e., mate detection. However, web-building spiders likely detect and use vibration differently than free-moving species like lycosids and others. Addressing these questions and others like this would significantly improve the manuscript and address its limitations and implications more clearly and thoroughly.  

Response 18: Repetition within these paragraphs has been removed. Further details and discussion are given in lines 351-362. We note that more research is currently underway to answer some of these questions - such as how far away the spider can feel the vibration (and different types of vibration) from its source, and how substrate affects this distance and their reaction, and how web-building spiders react to vibration compared to free-moving spiders.

Comment 19: Lines 310-313: The authors report data on sex and developmental stage differences in the discussion section when it should have been reported in the results, analyzed statistically when possible, and interpreted in the discussion.

 Response 19: Comments about the sex/gender and age of collected spiders have been removed as this is part of a different manuscript, focussing on these, currently being prepared for submission.

Congratulations again.

Reviewer 2 Report

Comments and Suggestions for Authors

This is a good paper and should be published. Biases associated with collection methods are always an issue with diversity studies, and so it is good to see a thorough comparison of different techniques. The vibration technique is apparently limited to spider studies, but overall the paper shows that different techniques do give different results wrt species diversity and overall collection success. Providing detailed costing at first seemed a little odd, but on reflection, diversity studies are often small studies with limited funding and so details of costing are useful. These costings may differ depending on where and when the studies are carried out, but the breakdown of necessary equipment and time taken for the various steps is more generally useful.

I did find some of the language a bit annoying (eg labelled “as such” where “accordingly” would be better), but these are issues for the editors.

Other points:

-       P4 “an arachnologist with over 40 years of experience.”- why not just name the person (RR I assume).

-       I like the Venn diagram but including % of spp. as well as number would be helpful. (yes, some of this info is given in text)

Table 1 is confusing:

-        survey method column – pls left align as looks like some are subsets of other

-        The footnote on the table doesn’t seem to have a corresponding “1” on the table itself, and I found the a b superscript system confusing.

Iine 184 “The vibration from the tractor emitted 184 G#maj13/C chord; a middle C which has a frequency of about 261.62 Hz.” This info belongs in the materials and methods.

Fig 3 would be a little clearer if each bar was labelled “species” and “family”, and the method written only once below the pair of bars.

Comments on the Quality of English Language

This is a good paper and should be published. Biases associated with collection methods are always an issue with diversity studies, and so it is good to see a thorough comparison of different techniques. The vibration technique is apparently limited to spider studies, but overall the paper shows that different techniques do give different results wrt species diversity and overall collection success. Providing detailed costing at first seemed a little odd, but on reflection, diversity studies are often small studies with limited funding and so details of costing are useful. These costings may differ depending on where and when the studies are carried out, but the breakdown of necessary equipment and time taken for the various steps is more generally useful.

I did find some of the language a bit annoying (eg labelled “as such” where “accordingly” would be better), but these are issues for the editors.

Other points:

-       P4 “an arachnologist with over 40 years of experience.”- why not just name the person (RR I assume).

-       I like the Venn diagram but including % of spp. as well as number would be helpful. (yes, some of this info is given in text)

Table 1 is confusing:

-        survey method column – pls left align as looks like some are subsets of other

-        The footnote on the table doesn’t seem to have a corresponding “1” on the table itself, and I found the a b superscript system confusing.

Iine 184 “The vibration from the tractor emitted 184 G#maj13/C chord; a middle C which has a frequency of about 261.62 Hz.” This info belongs in the materials and methods.

Fig 3 would be a little clearer if each bar was labelled “species” and “family”, and the method written only once below the pair of bars.

Author Response

This is a good paper and should be published. Biases associated with collection methods are always an issue with diversity studies, and so it is good to see a thorough comparison of different techniques. The vibration technique is apparently limited to spider studies, but overall the paper shows that different techniques do give different results wrt species diversity and overall collection success. Providing detailed costing at first seemed a little odd, but on reflection, diversity studies are often small studies with limited funding and so details of costing are useful. These costings may differ depending on where and when the studies are carried out, but the breakdown of necessary equipment and time taken for the various steps is more generally useful.

Comment 1: I did find some of the language a bit annoying (eg labelled “as such” where “accordingly” would be better), but these are issues for the editors.

Response 1: Where identified the text has been replaced as requested.

Other points:

Comment 2: P4 “an arachnologist with over 40 years of experience.”- why not just name the person (RR I assume).

Response 2: The name of the spider identifier, Dr Robert Raven, has been given in line 189.

Comment 3: -       I like the Venn diagram but including % of spp. as well as number would be helpful. (yes, some of this info is given in text)

Response 3: The percentages of each species have been included in the Venn diagram.

Table 1 is confusing:

Comment 4: survey method column – pls left align as looks like some are subsets of other

Response 4: The survey method column in Table 1 has been left aligned.

Comment 5: The footnote on the table doesn’t seem to have a corresponding “1” on the table itself, and I found the a b superscript system confusing.

Response 5: Within Table 1 “1” has been changed to ”ab”.  This is the standard method for easily identifying statistically significant differences between means in a Table.

Comment 6: Iine 184 “The vibration from the tractor emitted 184 G#maj13/C chord; a middle C which has a frequency of about 261.62 Hz.” This info belongs in the materials and methods.

Response 6: The line “The vibration from the tractor emitted 184 G#maj13/C chord; a middle C which has a frequency of about 261 Hz.” has been moved to lines 142-143.

Comment 7: Fig 3 would be a little clearer if each bar was labelled “species” and “family”, and the method written only once below the pair of bars.

Response 7: Figure 3 has been amended to the above advise.

Reviewer 3 Report

Comments and Suggestions for Authors

The authors don't factor in the cost of renting a tractor when considering how this method could be used for other researchers (they themselves were able to use one for free), which is bizarre since it is integral to their 'new' method. It is an assumption to think that every agricultural landowner will provide you with such equipment for these purposes. The method is also poor for collecting species richness, as shown in Table 1. Even pitfall traps, which are significantly biased to ground-dwelling species, were showing more success at species richness than the 'tractor method' although they take longer. Later in Line 252 they concede the traps were left for 6 weeks (!) which could indeed have allowed more specimens to accumulate. Perhaps, but even so, the data shows a clear pattern that the vibration method was not picking up species not already found easily in the night surveys. Some differences between pitfall and tractors for collecting females of Corinnidae etc. were shown, but rendered insignificant when one considers these can simply be picked up by night surveying instead.

No consideration of the environmental impact of running diesel engines has been considered anywhere in the paper, nor any comment on whether this may disturb other animals unnecessarily, such as herpteofauna. This will all have to be addressed if the authors want to uphold ethical standards. Pitfall traps have their own risks but those are clearly elucidated. The authors are biased in presenting the tractor surveying as better when they haven't considered the emissions and other possible implications as discussed above. In its present state, this paper is simply preliminary methodological work and some raw (albeit interesting) faunistic data. It does not significantly advance our knowledge in spider collection, something the editors should consider as Animals is supposed to be a journal for papers making strong theoretical contributions to the respective fields.

Line 139: "Spiders of all instars were identified by an arachnologist with over 40 years of experience." this is self-congratulatory, the amount of time someone has worked has no bearing on identification skills in a faunistic context, no real taxonomy was performed here. Identifying morphospecies is something undergraduate students can do, finding the correct nomenclature thereafter is what requires skill. Especially with juveniles, identification below the family level is often not possible in any case, without molecular sequencing. Identification to genus-level is possible in some families, but these are rather the exception.

Most problematic is the title "Vibration as a new survey method for spiders", other methods of vibration have been used for hundreds of years to collect spiders, such as tuning forks. In one brief check of the literature, I find no less than 3 other vibration methods which have been published on. Whilst the tractor is a new way of applying a vibration method, it is erroneous of the authors to suggest that they propose vibration itself as a new survey method for spiders. 

If they want this paper to be impactful, they need to address the above concerns, otherwise it is simply another faunistic paper with an methodological aside.

Author Response

Comments:

The authors don't factor in the cost of renting a tractor when considering how this method could be used for other researchers (they themselves were able to use one for free), which is bizarre since it is integral to their 'new' method. It is an assumption to think that every agricultural landowner will provide you with such equipment for these purposes. The method is also poor for collecting species richness, as shown in Table 1. Even pitfall traps, which are significantly biased to ground-dwelling species, were showing more success at species richness than the 'tractor method' although they take longer. Later in Line 252 they concede the traps were left for 6 weeks (!) which could indeed have allowed more specimens to accumulate. Perhaps, but even so, the data shows a clear pattern that the vibration method was not picking up species not already found easily in the night surveys. Some differences between pitfall and tractors for collecting females of Corinnidae etc. were shown, but rendered insignificant when one considers these can simply be picked up by night surveying instead.

No consideration of the environmental impact of running diesel engines has been considered anywhere in the paper, nor any comment on whether this may disturb other animals unnecessarily, such as herpteofauna. This will all have to be addressed if the authors want to uphold ethical standards. Pitfall traps have their own risks but those are clearly elucidated. The authors are biased in presenting the tractor surveying as better when they haven't considered the emissions and other possible implications as discussed above. In its present state, this paper is simply preliminary methodological work and some raw (albeit interesting) faunistic data. It does not significantly advance our knowledge in spider collection, something the editors should consider as Animals is supposed to be a journal for papers making strong theoretical contributions to the respective fields.

Line 139: "Spiders of all instars were identified by an arachnologist with over 40 years of experience." this is self-congratulatory, the amount of time someone has worked has no bearing on identification skills in a faunistic context, no real taxonomy was performed here. Identifying morphospecies is something undergraduate students can do, finding the correct nomenclature thereafter is what requires skill. Especially with juveniles, identification below the family level is often not possible in any case, without molecular sequencing. Identification to genus-level is possible in some families, but these are rather the exception.

Most problematic is the title "Vibration as a new survey method for spiders", other methods of vibration have been used for hundreds of years to collect spiders, such as tuning forks. In one brief check of the literature, I find no less than 3 other vibration methods which have been published on. Whilst the tractor is a new way of applying a vibration method, it is erroneous of the authors to suggest that they propose vibration itself as a new survey method for spiders. 

If they want this paper to be impactful, they need to address the above concerns, otherwise it is simply another faunistic paper with an methodological aside.

Response to Reviewer 3.

After two positive reviewers comments Reviewers 3 comments are somewhat strange and mostly difficult to respond to. 

In relation to the comments about renting a tractor – all of the Australian farmers (and the original idea actually came from a farmer) whom we have spoken to about this phenomenon are fascinated with the concept that an idling tractor can attract spiders and all have offered to contribute the use of a tractor for further research. Furthermore, as we made the point in the Conclusions ‘A more compact vibration generator needs to be developed to allow access to locations inaccessible to a tractor to survey spiders. When a smaller vibration generating device is developed, i.e. a tractor is no longer required, the vibration-based method could be a more time and cost-effective alternative to pitfall traps but should be used in conjunction with night collections to obtain a broader, more representative spider community assemblage.’ Thus, we were using the tractor to rigorously test the concept that vibration was effective in attracting multiple species of spiders by using a tractor under controlled (multiple locations, spiders systematically collected) and comparative (to other methods) situations.  In the future we can foresee use of a much smaller portable device to create the vibration, as mentioned in the Conclusions.

We acknowledge in the text that use of the vibration-based method resulted in fewer species collected than both night collections and pitfall traps, however we collected 24 species using vibration that were not collected during night collections and 42 species not collected in pitfall traps that were attracted to the vibration. The vibration-based method does have value by facilitating collection of species not otherwise seen, thus advancing our knowledge of spider taxonomy and collection, and with some evidence that it is more cost effective and timely.

Although we did not specifically discuss the environmental effects of running a diesel engine we did indicate that ‘a more compact vibration generator needs to be developed’ and thus we have no desire to incur the costs, emissions and environmental impact of repeatably using a tractor to survey spiders and indicate more environmentally friendly options need to be developed (and this research is underway).

Our initial advice was not to name the arachnologist; clearly this was an error and we have now done so (line 189). However, we find some of the comments e.g. ‘something undergraduate students can do’ and ‘identification below the family level is often not possible’ somewhat condescending and unprofessional. It may be true in other parts of the world (e.g. Europe) where their fauna is almost entirely known – in Australia the spider fauna has been estimated to be only 20-40% known, depending on which source you accept (see references below). Furthermore, a number of papers describing new species include juveniles in the ‘Material Examined…’.

It would have been useful for the reviewer to list the three papers ‘other methods of vibration have been used for hundreds of years to collect spiders, such as tuning forks’. The papers we have identified about the effects of vibration on spiders are more about the behaviour of individual spider species (mostly under laboratory conditions), certainly not on the scale of collecting dozens of spider species, systematically collected in multiple field locations, and at the same time compared to traditional methods.

References relevant to known spider fauna in Australia

40% in Raven, R. J. (1988). The current status of Australian spider systematics, pp. 37-47, Figs. 1, 2, 1 Table. In, Australian Arachnology, (eds.) Austin, A. D. and Heather, N.W., Australian Entomological Society, Miscellaneous Publ. No. 5, 137 pp.
No more than 20% according to Yeates, D. K., Harvey, M. S. & Austin, A. D. 2003. New estimates for terrestrial arthropod species-richness in Australia. Records of the South Australian Museum Monograph Series No. 7: 231–241.

Round 2

Reviewer 3 Report

Comments and Suggestions for Authors

It is disappointing to see the authors refused to make any of the suggested additions of caveats about the limitations of this study and the possible environmental impact of using a tractor. This would have improved the paper and takes only a few sentences if presented as preliminary comments. They have improved other areas of the manuscript based on the comments of other colleagues and these seem to have improved the paper somewhat.